

# Histological analysis of post-eruption tooth wear adaptations, and ontogenetic changes in tooth implantation in the acrodontan squamate *Pogona vitticeps*

Yara Haridy

Department of Biology, University of Toronto Mississauga, Mississauga, Ontario, Canada
Museum für Naturkunde, Leibniz Institute for Evolution and Biodiversity Science, Invalidenstraße, Berlin, Germany

## ABSTRACT

Teeth have been a focus of research in both extinct and extant taxa alike; a significant portion of dental literature is concerned with dental patterning and replacement. Most non-mammalian vertebrates continuously replace their dentition but an anomalous group of squamates has forgone this process in only having one tooth generation; these squamates all have apically implanted teeth, a condition known as acrodonty. Acrodont dentition and various characteristics attributed to it, including a lack of replacement, have often been defined ambiguously. This study explores this type of implantation through histology in the ontogeny of the acrodont agamid *Pogona vitticeps.* The non-replacing teeth of this squamate provides an opportunity to study wear adaptations, maintenance of occlusion in a non-mammalian system, and most importantly post-eruption changes in the tooth bone interface. In this study the post-eruption changes combined with dental wear likely gives the appearance of acrodont implantation.

## INTRODUCTION

Reptilian dentition has been extensively studied in both extinct and extant taxa, and for the vast majority of these taxa, there is constant replacement of teeth, a condition known as polyphyodonty. However, in a subset of reptiles, there has been an evolutionary cessation of replacement, a condition known as monophyodonty. Among lepidosaurian reptiles, this suppression of replacement is limited to Sphenodontidae (Rhynchocephalia), Chamaeleonidae, and Agamidae, with the latter two being grouped within the clade Acrodonta (Squamata) (*Pyron, Burbrink & Wiens, 2013*). This squamate group is aptly named for the acrodont style implantation of the dentition, and all acrodontians have apically implanted teeth making up the majority of their dentulous surface (*Edmund, 1960*; *Peyer, 1968*; *Zaher & Rippel, 1999*; *Cooper, Poole & Lawson, 1970*; *Jenkins et al., 2017*). This is in contrast to the condition found in most squamates and other reptiles (e.g., *Zaher & Rippel, 1999*; *Delgado, Davit-Beal & Sire, 2003*; *LeBlanc & Reisz, 2015*) in which the tooth is implanted to the lingual surface of the jaw bone, a condition known as pleurodonty that

Corresponding author
Yara Haridy, yara.haridy@mfn.berlin, yara.haridy@mail.utoronto.ca

is exemplified in taxa like *Iguana iguana* (*Throckmorton, 1976*; *Montanucci, 2008*; *Kline & Cullum, 2017*). The most studied form of implantation among tetrapods is thecodonty, where the tooth is implanted in a deep socket; this form of implantation is found in all mammals and also occurs within crocodilians and in many extinct archosaurs (e.g., *Zaher & Rippel, 1999*; *Brown et al., 2015*; *García & Zurriaguz, 2016*). Tooth implantation should not be conflated with tooth attachment, which refers to the tissue that attaches the tooth to the dentulous bone. This study is primarily concerned with implantation, and how ontogenetic change can influence the appearance of implantation categories.

True teeth in most non-mammalian vertebrates are well known for their extensive replacement patterns through their life (*Zaher & Rippel, 1999*; *Delgado, Davit-Beal & Sire, 2003*; *LeBlanc & Reisz, 2015*; *LeBlanc et al., 2016b*); therefore, any condition that deviates from polyphyodonty is unusual and worthy of study. An important example of non-polyphyodont dentition is seen in mammals, in which there are only two generations of dentition, the deciduous teeth and the permanent teeth (*Sire et al., 2002*). Some mammals such as shrews have even forgone the deciduous phase by resorbing the dentition prior to eruption, functionally giving them one tooth generation (*Järvinen, Tummers & Thesleff, 2009*). Acrodonty in Reptilia is largely associated with a lack of tooth replacement (e.g., *Zaher & Rippel, 1999*; *Smirina & Ananjeva, 2007*), an association that often alludes to causality but that is never stated outright. At first glance it seems understandable that acrodont implantation and monophyodonty be associated this is because most acrodont squamates have both acrodont and pleurodont dentition and in individuals with both pleurodont and acrodont dentitions, the pleurodont teeth are replaced, yet the acrodont teeth are not (*Cooper, Poole & Lawson, 1970*). This discrepancy seems to have cemented the idea that acrodonty somehow inherently disrupts or inhibits replacement. While extant acrodont squamates are monophyodont, it is important to note that acrodonty is also found in some extinct reptiles (*Simões et al., 2015*; *Haridy, LeBlanc & Reisz, 2018*) and in non-reptilian vertebrates such as piranhas (*Shellis & Berkovitz, 1976*), all of which replace their dentition.

The supposed function of constant replacement is to avoid excessive wear (*Throckmorton, 1979*; *Benton, 1984*; *Erickson, 1996*); accordingly, higher rates of replacement are often seen in herbivorous taxa, as their fibrous diet and frequency of mastication requires a constant renewal of their dentition. In extreme instances, there are examples of starvation in herbivorous mammals that lack continuous replacement and that have worn their dentition to such a degree such that it is no longer functional (*Spencer, 2005*). Therefore the lack of replacement in acrodont squamates raises the question: if acrodontians that lack tooth replacement are relatively long-lived (*Zari, 1999*; *Smirina & Ananjeva, 2017*), how are they able to maintain a viable occlusal surface during later stages of life? More specifically: (1) how do acrodontians adapt to dental wear at a tissue level, and (2) how do the mandibular and maxillary teeth stay in occlusion through ontogeny without replacement or a ligamentous attachment? The former of the two questions has been partially addressed through histology of the acrodontians *Uromastyx aegyptia* (*Throckmorton, 1979*) and *Chamaeleo calyptratus* (*Buchtová et al., 2013*; *Dosedělová et al., 2016*), and members of acrodontan Uromastycinae and Chamaeleoninae (*Pyron, Burbrink & Wiens, 2013*). These

studies found that these acrodontians have peculiar way to combat wear—they secondarily infill the teeth with bone and/or dentine and remodel the underlying jaw bone into more compact bone. There are six other groups within Acrodonta that presumably also lack replacement and have undocumented wear adaptations (*Pyron, Burbrink & Wiens, 2013*). This study tests if the agamid *Pogona vitticeps* (the central bearded dragon) conforms at a tissue level to the wear adaptation patterns of other acrodontians previously documented in literature.

To better understand the relationship between acrodonty, monophyodonty, and wear adaptations, the implantation mode itself should be more thoroughly examined, particularly when it comes to acrodonty it is the least studied form of implantation. Acrodont implantation has been ambiguously defined at best, with various authors describing the mode of implantation as: (1) apically placed teeth (*Edmund, 1960*), (2) teeth ankylosed to the margin of the jaw (*Motani, 1997*), (3) teeth fused to the edge of the jaw bone via undefined tissues (*Peyer, 1968*). Most definitions used acrodontian squamates or rhynchocephalians as representative groups for acrodont implantation, with acrodonty used as a phylogenetic character (e.g., *Zaher & Rippel, 1999*). In this paper we use the most general definition of acrodonty as established by *Edmund (1960)*, referring only to the apical position of the teeth in relation to the jaw.

The classic categories of acrodonty, pleurodonty, and thecodonty have been used as descriptors, as well as phylogenetic characters (*Zaher & Rippel, 1999*), and continue in even the latest literature (*Jenkins et al., 2017*) but have been called into question by *Estes & Charles (1988)* who suggested these categories are artificial and are not likely representative of natural groupings. To ground truth the use of acrodontians as acrodont representatives the natural question becomes: are the acrodont teeth of acrodontians really acrodont? It was proposed that ichthyosaurs and alligators change implantation types ontogenetically by growth in the jaw ramus and additional ossification on the lingual side (*Motani, 1997*), alligators go from pleurodont to thecodont, and the ichthyosaurs become subthecodont from their juvenile state of pleurodonty. This study explores the ontogenetic change in the teeth of *Pogona vitticeps,* with a focus on comparisons of wear adaptations of this member of Agamidinae to those previously described in Uromastycinae and Chamaeleoninae. This study aims to explain: (1) Do tissue level wear adaptations documented in other acrodontians extend to *P. vitticeps*? (2) How do monophyodont reptiles that lack a ligamentous attachment maintain occlusion through ontogeny? (3) Are the teeth of these reptiles truly acrodont, and does acrodonty limit tooth replacement as indicated by literature? This is the first study to document the change in implantation types through ontogeny due to osteological remodeling of the dentary and the tooth body, with evidence of a change from pleurodont implantation to acrodont implantation in a modern squamate. This has implications on how we view implantation categories, as they are likely to be more ontogenetically variable than previously thought.

## MATERIALS AND METHODS

In this study, the genus *Pogona* was represented by the central bearded dragon (*P. vitticeps*). Thirty-seven specimens (Fig. S1) of *P. vitticeps* from the Royal Ontario Museum (ROM)

recent osteology collection were externally examined, and all measurements were taken prior to skeletonizing which allowed the species to be identified. Specimens sectioned here were captive bred and donated to the ROM. All specimens were photographed using a Canon EOS40D prior to sectioning. To access normally inaccessible anatomical features of the lower jaw and its dentition, five specimens were sectioned in both coronal and longitudinal planes. *Pogona vitticeps* was sectioned at two ontogenetic stages, a juvenile stage (ROM R8234, ROM R8510), and an adult stage (ROM R8507, ROM R8189, ROM R9422). The illustrations and diagrams found in the figures were made using Adobe Photoshop CS6 and Adobe Illustrator CS6.

## Histology

All thin sections were made following the ROM histology protocol and were executed in the ROM vertebrate paleontology thin sectioning facility. Specimens were embedded in AP Castolite acrylic resin, vacuumed, and left to cure for a minimum of 24 h. All specimens were cut using a Buhler Isomet 1000 wafer saw at a low speed of 275 rpm. The specimens were mounted on plexiglass slides using Scotch-Weld SF-100 cyanoacrylate. The slides were then mounted on the Hillquist grinding cup and ground down using the grinding cup until optical clarity was achieved; subsequently the specimen was manually ground using progressively finer grit suspensions on glass plates, beginning with a 600-grit silicon carbide powder and working down to a 1-micron aluminum oxide powder. All slides were imaged using a Nikon DS-Fi1 camera mounted to a Nikon AZ 100 microscope fitted with crossed-polarizing and lambda filters and an oblique illumination slider and NIS-Elements software registered to R. R. Reisz of the University of Toronto Mississauga.

## RESULTS

### External anatomy

The external morphology of both the mandible and dentition is noticeably different between the juvenile and adult specimens of *Pogona vitticeps* (Figs. 1A, 1B). Several specimens that represented a relatively complete ontogenetic sequence were examined (see Table S1); for the sake of brevity, two representative specimens, an adult (ROM R8507) and a juvenile (ROM R8234), were chosen for external description. The juvenile specimen (ROM R8234) has fewer tooth positions (16 tooth positions) on the dentary; however, smaller individuals were found to have as few as 10 teeth. There are external mandibular features of ROM R8234 that identify it as juvenile: (1) the symphysis is poorly ossified, (2) the mandible is less robust in bone density, which is externally apparent as a character of bone opacity, and in dorsoventral width, and (3) the mandible is relatively short, with the dentulous region making up more than half the total length. The wear facets on the labial surface of the dentary are not as well developed in the juvenile specimen as those seen in the adults. This specimen of *P. vitticeps* is found to have two teeth implanted in a pleurodont fashion; these occupy the first two tooth positions on the rostral end of the mandible (Fig. 1B). The pleurodont teeth are conical and taper to a point; this represents the most common condition among many, but not all agamids (*Cooper, Poole & Lawson, 1970*). Posterior to the pleurodont dentition, the remaining tooth row has been described in literature to

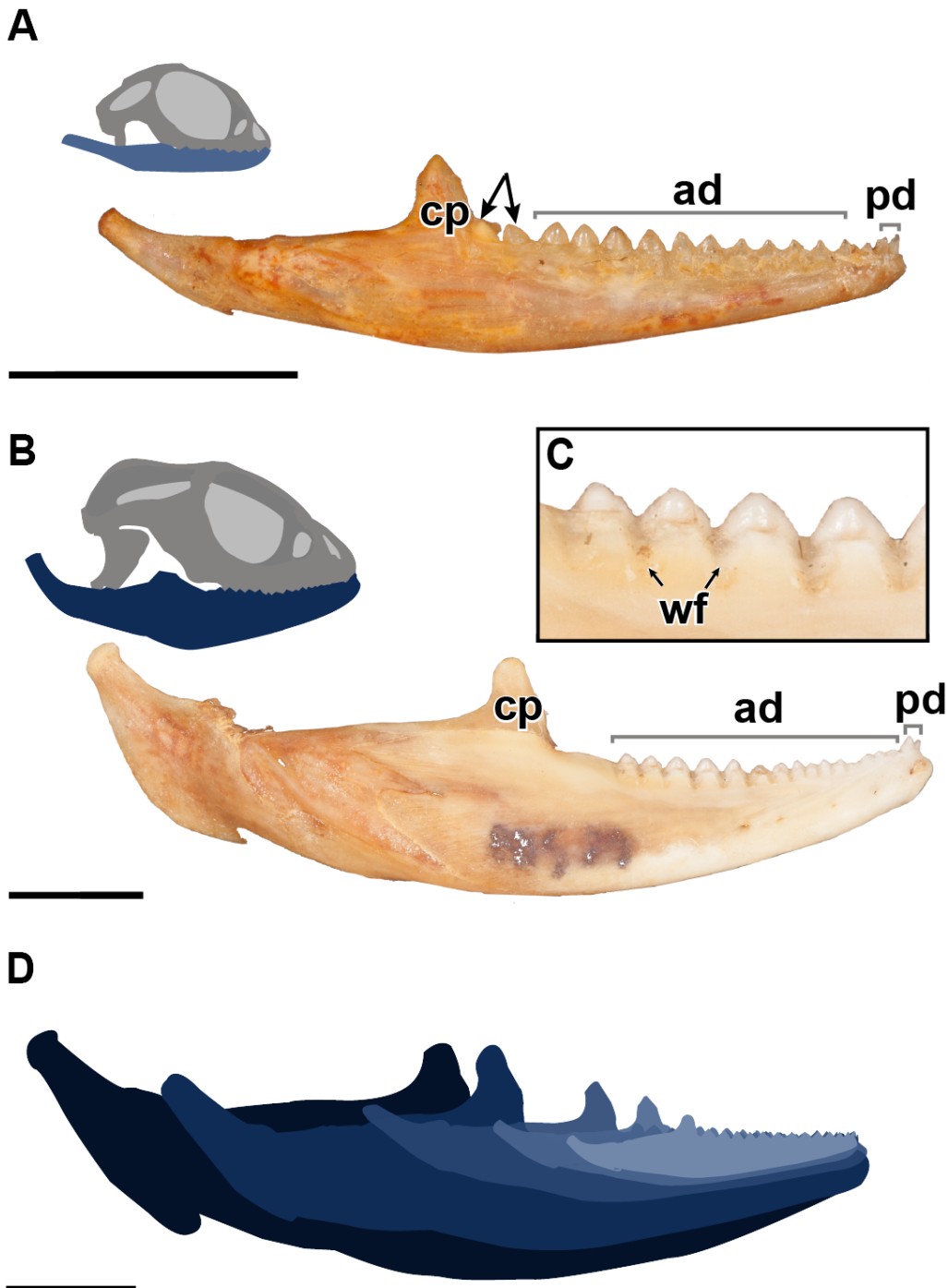

**Figure 1** **A comparative figure showing the external morphological differences in the dentition and mandibles between juvenile and adult specimens of *P. vitticeps*.** (A) Right mandibular ramus of juvenile specimen ROM R8234; (B) Right mandibular ramus of adult specimen ROM R8507;(C) Closeup of wear facets on dentary in adult *P.vitticeps* (D) superimposed outlines of mandibles of varying ontogenetic stages, showing that most growth occurs in the posterior end of the mandible. Abbreviations; ad, acrodont dentition; cp, coronoid process; pd, pleurodont dentition; wf, wear facets. Arrows indicating un-ankylosed teeth. Scale bar = 1 cm.

be acrodont and is the focus of this study (*Cooper, Poole & Lawson, 1970*). The acrodont teeth in *P. vitticeps* are triangular and mediolaterally compressed and lack the multi-cuspid morphology that other acrodontians (Fig. S1), like chameleons and *Uromastyx* display. Compared to the older anterior teeth, the teeth are larger in size posteriorly correlating with age, with the posterior teeth being the largest and the youngest. It appears that the newest tooth is not ankylosed to the jaw bone prior to the development of the next tooth in the series. Instead, the newest tooth appears be attached to the jaw only by soft tissue. Desiccated fibrous tissue separates it from the bone in skeletonized material, (Fig. 2); this is readily recognizable by external observation, because the last two teeth are not oriented at an angle that is congruent with the rest of the dentition (Fig. 1A). This was also observed in several other juvenile specimens in which it is observed only in the posteriormost teeth. Lastly, the newest and last tooth position is growing directly against, and partially resorbing (Fig. 1A), the coronoid process, which does not allow sufficient space for the next more posterior tooth position.

As with the juveniles, several adult individuals were examined, and a representative specimen chosen. Several changes correlated with size and thus presumably with ontogeny can be recognized in the adult specimen of *P. vitticeps* (ROM R8507) including an increase in the number of tooth positions. The specimen chosen for sectioning has 17 tooth positions, although other specimens have been found to have as many as 19 tooth positions. In most specimens a definite tooth count was difficult to ascertain via examination of the external morphology due to extensive wear but was later verified in thin section. Wear on the adult mandible is evident on both the acrodont dentition and the jaw bone. Interestingly, the anterior pleurodont dentition was mostly unworn in the adult specimen or worn to a minimal degree on other adult specimens examined; this is similar to the condition seen in *Agama agama*, which has been documented to replace its anterior pleurodont dentition (*Cooper, Poole & Lawson, 1970*). However, the anterior acrodont teeth are almost completly worn away, making it difficult to differentiate it from the jaw bone; to circumvent this problem, tooth counts were made under a microscope and later confirmed in thin section when possible. The wear observed on the anterior acrodont dentition is extensive—often there were only traces of the tooth left—which is similar to the dental wear seen in *Uromastyx*, a taxon that can become functionally edentulous in adulthood (*Throckmorton, 1979*).

The evidence of extensive wear is not only seen on the teeth but is also found on the mandible itself in the form of wear facets, which are only present on the labial side of the dentary where the maxillary dentition occludes, interdigitating between dentary tooth positions. Wear facets have been characteristically found on the mandibles of acrodont squamates and rhynchocephalians as far back as the Cretaceous (*Simões et al., 2015*), which are formed due to the maxillary dentition wearing down on the dentary bone during mastication and passive occlusion. These wear facets are present along the posterior two-thirds of the mandible but are most developed in the posterior region of the dentary of adult specimens of *P. vitticeps.* However, in adult individuals of other acrodontians, like chameleons (*Dosedělová et al., 2016*), the wear facets are found along the entire dentary. Lastly, it is important to mention that the jaw ontogenetically increases posteriorly in

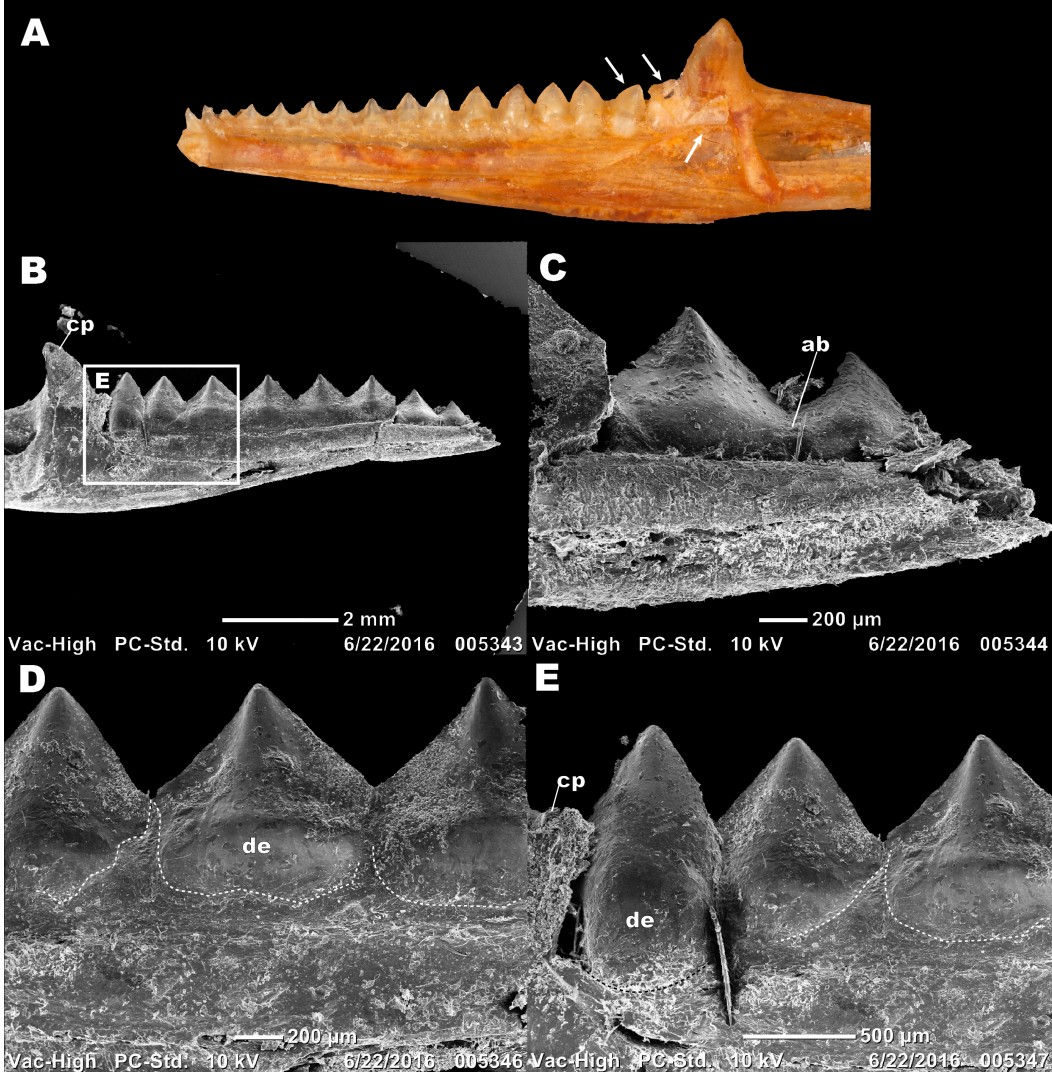

**Figure 2** **Details of external morphology of the juvenile dentition of *P. vitticeps*.** (A) ROM R8234 in lingual view showing un-ankylosed dentition, the most posterior growing into the coronoid; (B) ROM R8418 scanning electron microscope image of incomplete juvenile mandible with; (C) closeup showing fibrous alveolar bone between the teeth; (D) closeup showing the lingual contribution of dentine to the teeth; (E) closeup showing the youngest tooth growing into and resorbing the coronoid. Abbreviations; ab, location of alveolar bone; cp, coronoid process; de, dentine. Arrows showing un-ankylosed teeth.

length and dorsoventrally in width, with the articular increasing in robustness, and the coronoid process moving posteriorly relative to the tooth row. This ontogenetic change effectively creates more space for the posterior addition of teeth to the dentary, a feature that is frequently seen in lepidosaurs (*Berkovitz & Shellis, 2016* and referances therein).

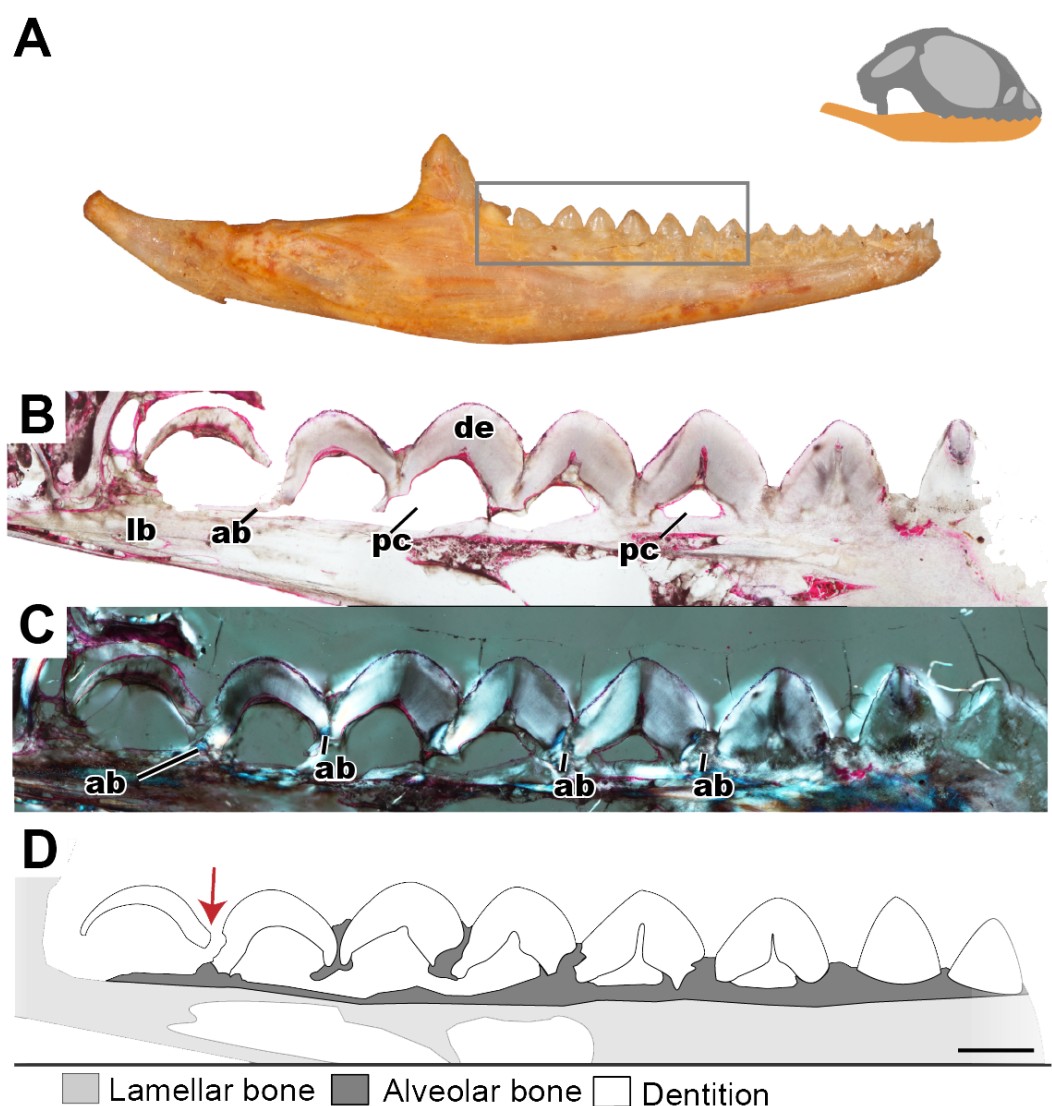

**Figure 3** **Longitudinal histological sections of the juvenile mandible of *P. vitticeps* with a focus on the dentition.** (A) External view of ROM R8234, box outlining the dentition cut; (B) a section of the posteriormost seven tooth positions. (C) a section of the posterior seven tooth positions in cross polarized light; (D) a schematic representation showing the distinct tissues as well as the progressive ankylosis of the teeth. Abbreviations; ab, alveolar bone; de, dentine; lb, lamallar bone; pc, pulp cavity. Red arrow indicates where the newest tooth is resorbing the previous tooth. Scale bar = 1,000 um.

## Histology

### Juvenile tissue histology

Longitudinal and coronal sections of the mandible of the juvenile specimen (ROM R8234) were examined (Figs. 2 and 3). The mandibular ramus in the juvenile specimen is not as well ossified as in the adult (ROM R8507) and has a medial curvature anteriorly; this made obtaining sections of the anterior and posterior dentition within the same sectioning plane difficult. Furthermore, the focus of this study is the non-replacing acrodont dentition, and

therefore, the anterior pleurodont dentition were not included in this study. In longitudinal section, only the dentine is exposed with no other dental tissues (e.g., cementum, enamel) being visible, this is likely due to sectioning bias. In longitudinal section, the implantation relationship of the teeth to the jawbone is not easily discernable. However, this plane of section allows for visualization of the incremental maturation of dentition. The most anterior teeth are the oldest, a characterization based on the thickness of dentine, whilst the most posterior teeth are the youngest and have the least amount of dentine infilling. In longitudinal section, the interaction between the individual tooth and its neighboring teeth can be seen in this plane (Figs. 2B–2D); there is distinct tissue that attaches the teeth to each other as well as to the jaw bone. This tissue is tentatively identified as alveolar bone (*sensu LeBlanc & Reisz, 2015*) based on the position and presumed function of the tissue rather than on its histological appearance, which does not completely conform to alveolar bone description in literature. The alveolar bone in this specimen has a woven appearance that is less organized than the bone that makes up the dentary. However, it lacks the extensive trabecular or 'spongy' appearance that is found in other squamates, such as snakes (*Budney, 2004*; *Budney, Caldwell & Albino, 2006*). This tissue attaches the teeth to each other, as well as to the jaw bone proper, a function that has been attributed to 'bone of attachment' (*Ananjeva & Smirina, 2007*), better known as alveolar bone (*Budney, Caldwell & Albino, 2006*; *Caldwell, 2007*; *LeBlanc & Reisz, 2013*; *LeBlanc & Reisz, 2015*; *LeBlanc et al., 2016a*). The alveolar bone in *P. vitticeps* is also not distinguished from the jaw bone by a reversal line; this is possibly an effect of the plane of section or due to a lack of resorption prior to attachment (Fig. 2). It is important to note that not all the teeth in the juvenile specimen are fully ankylosed to the dentary (Fig. 3); however, each tooth seems to ossify to the adjacent dentition, which occurs prior to full ankylosis and which mirrors what has been documented in *Chamaeleo calyptratus* (*Buchtová et al., 2013*; *Dosedělová et al., 2016*). This observation is based on the youngest tooth (Fig. 3), which has yet to ankylose to the dentary, which is attached to the neighboring teeth.

In coronal section, all of the common dental tissues are identifiable, with dentine comprising the bulk of the tooth identified by its characteristic radiating dentinal tubules. The enamel is best visualized in cross-polarized light and is unworn in the youngest teeth (Fig. 4B). Because of the mediolateral compression of the tooth, the coronal plane of section is the best plane in which to examine the enamel as it transects the labial and lingual surfaces of the tooth this. The enamel is fairly thick, which has also been reported in *Uromastyx* (*Throckmorton, 1979*). Due to the lack of wear in the juvenile specimen, the enamel is of equal thickness on the lingual and labial sides of the tooth crown (Fig. 4B). On the lingual side of the tooth, the tapering edge of the enamel leads down to a layer of acellular cementum, meeting at the cementoenamel junction, which defines the boundary between the anatomical crown and root of the tooth. It is likely that the cementum cannot be visualized on the labial side because the alveolar bone has grown to meet the cementum, effectively ankylosing the tooth while obstructing the cementum, or possibly because of resorption of the cementum in this process, although the latter has not been previously documented.

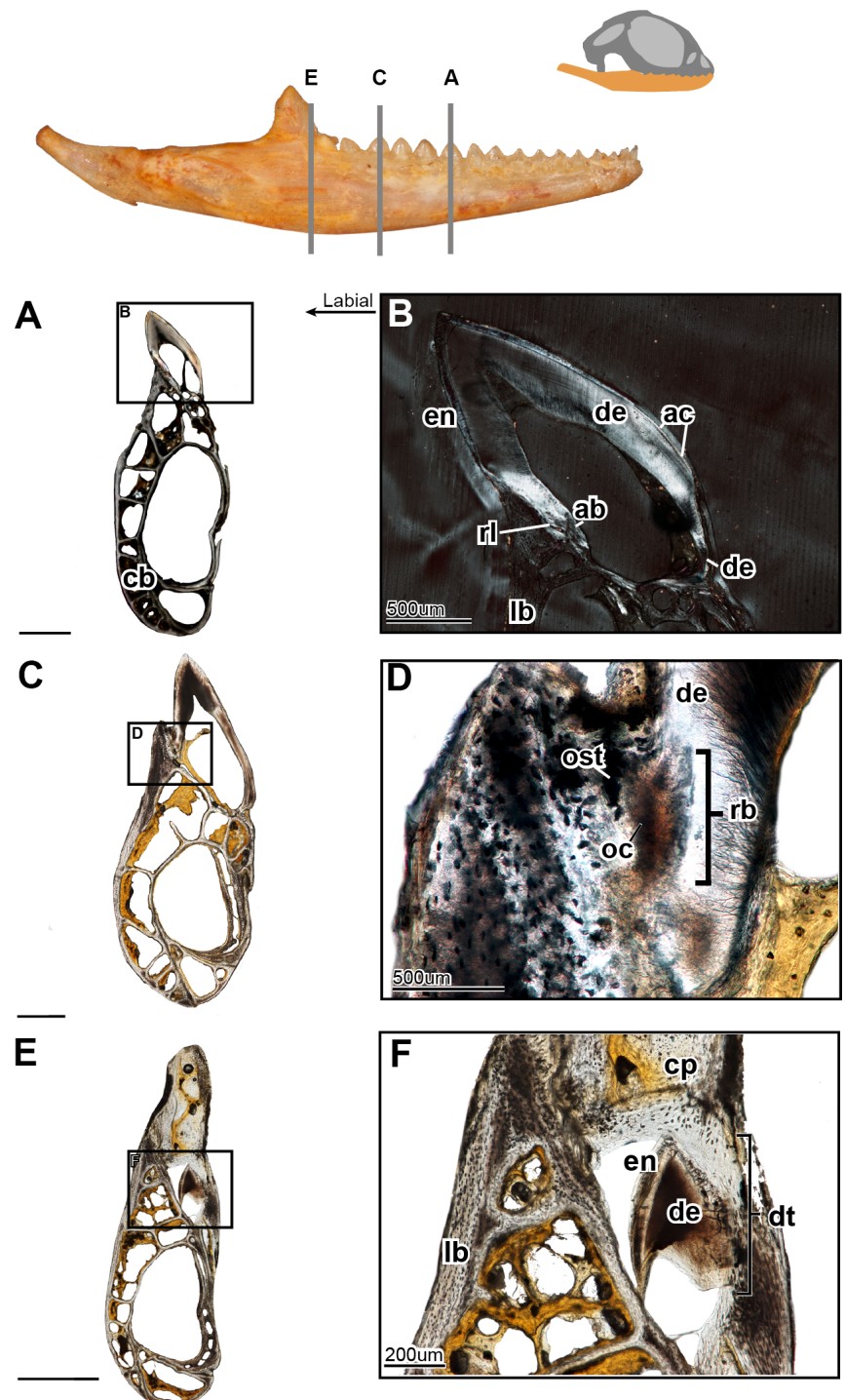

**Figure 4 Coronal sections of juvenile mandible of *P. vitticeps* with a special focus on tooth histology.**
(A) Coronal section of the jaw and tooth of a juvenile specimen of *P. vitticeps* ROM R8510 showing min-imal ossification of the jaw bone, and pleurodont tooth attachment; (B) close up of a juvenile specimen's (ROM R8510) dentition, showing unworn morphology; (C) (continued on next page...)

**Figure 4 (…continued)**
coronal section of a juvenile specimen (ROM R8234) showing pleurodont implantation, and remodeling; (D) closeup of the attachment site in ROM R8234, showing the labial side of the dentition being resorbed by an osteoclast; (E) coronal section of the coronoid process (ROM R8234) showing a tooth developing lingually inside the coronoid and dentary; (F) a close up of the tooth in the jaw bone, showing the resorption of the jawbone around newly developing tooth. Abbreviations: ab, alveolar bone; ac, acellular cementum; cb, cancellous bone; cp, coronoid process de, dentine; dt, developing tooth; lb, lamellar bone; en, enamel; ost, osteocyte lacunae; oc, osteoclast; rb, resorption bay; rl, reversal line. All un-labeled scale bars = 1 mm.

In the coronal section of the juvenile specimen of *P. vitticeps*, it is evident that both the shape of the dentition and the manner in which the tooth is implanted are more congruent with the pleurodont condition, as the lingual side of the tooth is markedly greater in length than that of the labial side (Figs. 4A, 4C). Although both ends of the tooth are ankylosed to the jaw, it is clear that there is more contact with the jaw bone and more attachment tissues on the labial side, a character of pleurodont implantation. Apart from the obvious labial bias of attachment, alveolar bone can be seen at the base of the tooth, being more woven than the rest of the jaw bone yet apparently lacking the more porous structure that is usually associated with alveolar bone. In coronal section (Fig. 4B), the reversal line defining the boundary between the new alveolar bone and the preexisting jaw bone is visible in cross-polarized light. The labial side of the tooth (Fig. 4C) is shorter and is attached to the jaw bone with more attachment tissue than that present at lingual side; furthermore, the labial side appears to be actively remodeled by osteoblasts and osteoclasts (Fig. 4D). The osteoclast on the labial side of the tooth is identified on the basis of its large size, general shape and the resorption bay that is created in the dentine (*Witten & Huysseune, 2009*). The high density of bone cell lacunae directly posterior to the osteoclast are identified as osteocytes, which were osteoblasts, indicating that the labial side of the tooth is being resorbed and that bone is being deposited in its place. It is also important to note that the strut like structure of the cancellous bone likely give the mandible form and function prior to full ossification and maturity.

Lastly, a coronal section through the coronoid process (Fig. 4E), shows the presence of a developing tooth. Its identification as a developing tooth is based on its general shape, location within the jaw, and the presence of thick enamel relative to the amount of dentine. The presence of more enamel than dentine denotes an early stage of tooth development, as dentine is deposited by odontoblasts later in development (*Erickson, 1996*; *LeBlanc et al., 2016b*). The presence of this developing tooth was serendipitously discovered when sectioning the specimen, and there were no external indicators that this tooth was buried within the bone of the coronoid process. This new tooth was developing within the coronoid process and was visibly resorbing the bone tissues of both the coronoid and the dentary, effectively making space for itself prior to attachment.

### Adult tissue histology

In the longitudinal and coronal sections of the mandible of the adult specimen (ROM R8057), lamellar bone makes up the main body of the dentary (Fig. 5). In coronal section, it can be seen that the large trabeculae seen in the juvenile have been incrementally infilled

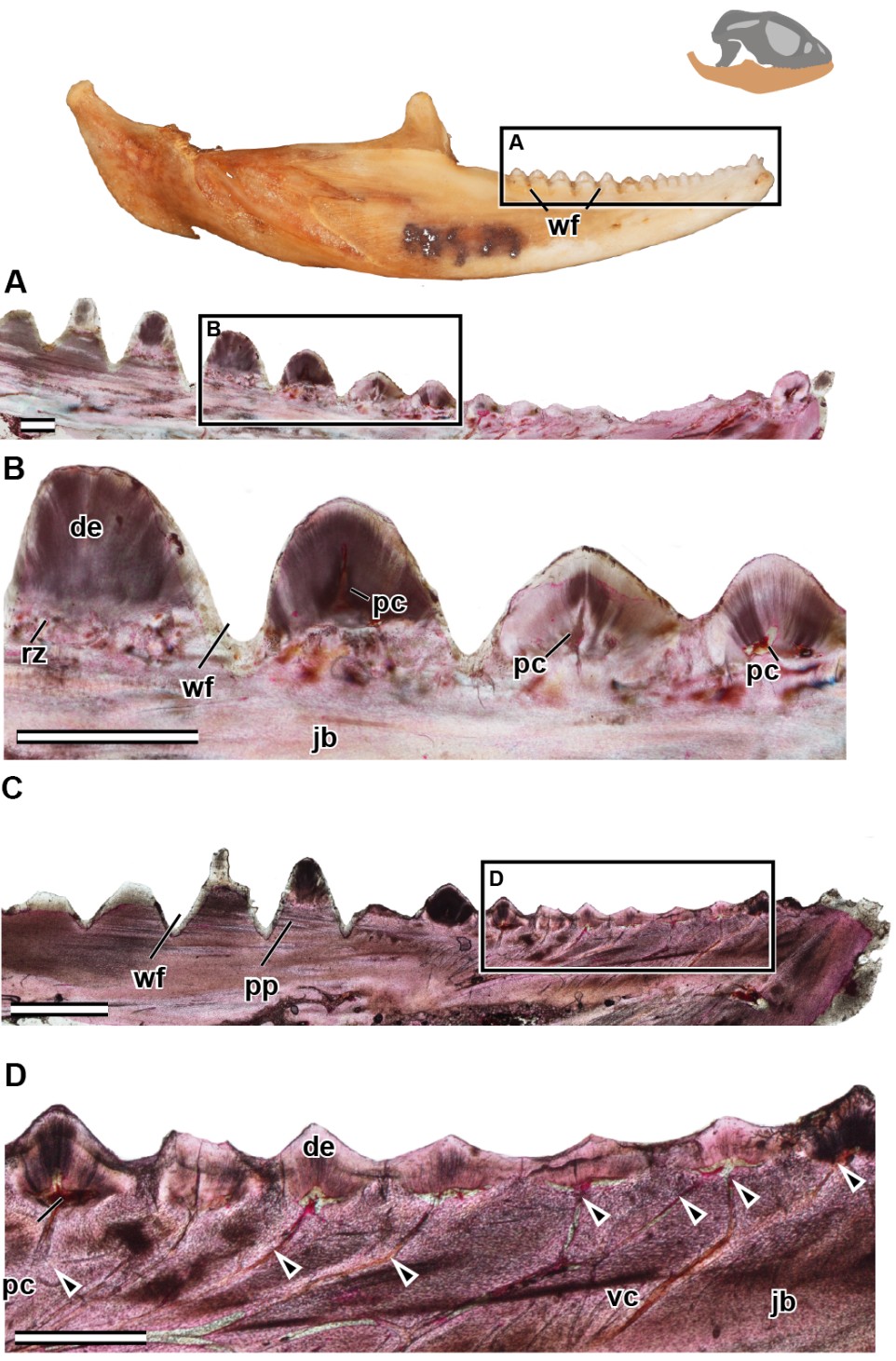

**Figure 5  Longitudinal sections of adult mandible of *P. vitticeps* with a focus on tooth histology.** (A) External view of specimen ROM R8507, box outlining the dentulous area sectioned; (B) broad view showing the variation between the anterior and the posterior *(continued on next page...)*

 

**Figure 5 (…continued)**
dentition; (C) a closeup of the worn dentition showing wear facets and remodeling zones; (D) broad view
showing the variation between the anterior and the posterior dentition, showing the depth of wear facets
and arrangement of vasculature; (E) a close up of the anterior dentition with open pulp cavities and asso-
ciated vasculature. Abbreviations: de, dentine; jb, jaw bone; pc, pulp cavity; pp, pseudo-pedicle; vc, vascu-
lar canal; wf, wear facet. The arrows demarcate the extensive vasculature leading to pulp cavities. Scale bar
= 1,000 um.

by lamellar bone, giving the mandible an osteological density that was not present in early ontogeny (Fig. 6). As previously reported in agamids (*Ananjeva & Smirina, 2007*), the bone tissue is not highly vascularized nor actively remodeled, particularly in dentary bone. This is in line with the findings in the adult jaw, where no reversal lines or large areas of remodeling are found. No primary or secondary osteons can be identified in the longitudinal sections. In longitudinal section (Fig. 5C), posterior to the most worn dentition, the wear facets become a marked feature of the dentary's labial surface. The pervasiveness and the depth of the wear facets should not be considered informative in thin section, as the variation of depth is a false impression, and is due to the sectioning plane in combination with the curvature of the dentary. The wear facets form pseudo-pedicels for the tooth remnants; these should not be confused for the 'bony pedicles' that have been identified developmentally in chameleons (*Buchtová et al., 2013*). The 'bony pedicles' in chamelons are formed by bone growing upwards to meet the developing tooth whilst the pseudo-pedicels in *P. vitticeps* are formed by the wear facets on the mandible, as the maxillary teeth wear away lamellar bone between the functional dentition on the jaw, leaving the remaining tooth caps on secondarily formed pedicels of lamellar bone (Figs. 5C–5D). This gives the tooth implantation region its distinctly acrodont appearance.

The acrodont teeth in the adult specimen are markedly different than those described in the juvenile (Fig. 6). The unequal shape and implantation of the teeth reported in the juvenile is not recognized in the adult, which has acrodont implantation with no lingual bias. The anterior pleurodont dentition is shown to maintain its vasculature, which is associated with an open pulp cavity; this is typical of pleurodont dentition among squamates. Perhaps the most interesting feature of the longitudinal sections is seen in the worn acrodont teeth directly posterior to the pleurodont pair (Fig. 5). The teeth are worn to such an extent that often little dentine and no enamel is detectable (Fig. 5B), even in thin section. However, the vasculature is maintained and denotes tooth positions, and the pulp cavity remains open and vascularized throughout the functional life of the tooth (Figs. 5 and 6). Thus, the teeth appear to remain viable into adulthood, which is in contrast to the condition reported in *Chameleo* (*Dosedělová et al., 2016*) and in *Uromastyx* (*Throckmorton, 1979*).

Considering the high degree of post-ankylosis changes to the tooth-bone interface, it is important to characterize the tissues involved. In the adult specimen, the tissues are distinct between the tooth and the platform of lamellar bone. The presence of these tissues alone provides evidence of remodeling, as they are not seen in the juvenile specimen (Fig. 4) and some tissue which were present no longer are identifiable in the adults (Fig. 6) In this case, 'remodeling' is used in the broad sense rather than to refer strictly to bone remodeling. For

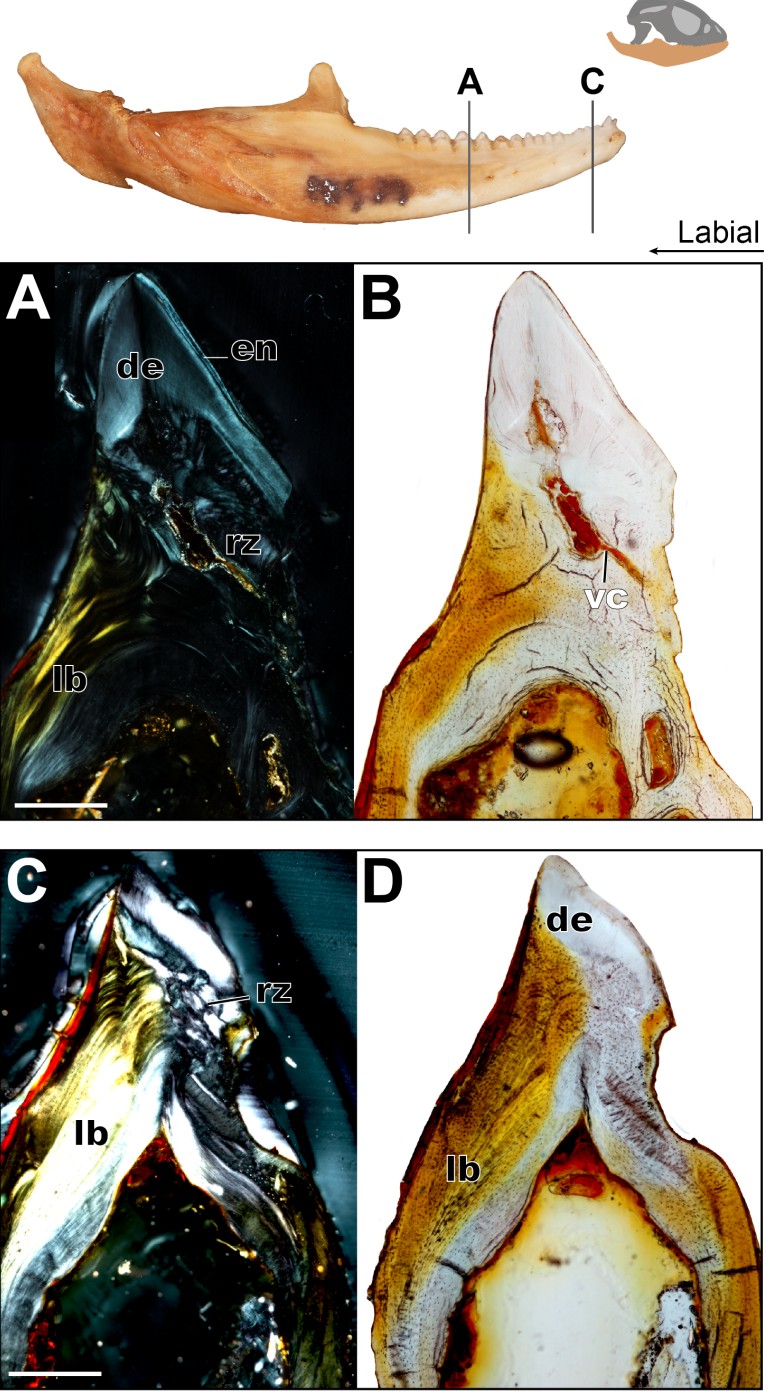

**Figure 6  Coronal sections of adult mandible of *P. vitticeps* with a focus on tooth histology.** (A) Coronal section of the jaw and tooth of an adult specimen of *P. vitticeps* (ROM R8507) showing the maintained vasculature and worn enamel on the labial side, in cross polarized light; (B) Coronal section of 'A' in normal transmitted light; (C) Coronal section of the jaw and tooth of an adult specimen of *P. vitticeps* ROM R8507 showing the extensive remodeling of the tooth attachment site as well as the tooth, in cross polarized light; (D) Coronal section of 'C' in normal transmitted light. Abbreviations: de, dentine; lb, lamellar bone; en, enamel; vc, vascular canals; rz, remodeling zone. Scale bar = 500 um.

example, in this agamid, the dentine that was once present in the lingual side of the tooth of the juvenile has been resorbed and is no longer present, or it is present in a greatly reduced extent in the adult. In both instances, it has been replaced by woven bone. Positioned between the dentine and the organized lamellar bone of the jaw is a layer of woven bone can be seen at the base of the teeth in longitudinal section (Fig. 5). The woven appearance under cross-polarized light identifies this as woven bone, which is characterized by the unorganized fibers (*Padian & Lamm, 2013*) (Fig. 6) the woven bone lacks vascular spaces, which are also mostly missing in the lamellar bone. The lack of organization in this bone indicates the occurrence of relatively fast deposition and is unlikely to be the original alveolar bone that was identified in the juvenile specimen (*Hernandez, Majeska & Schaffler, 2004*). The true extent of remodeling and change in tooth morphology is best seen in cross-polarized light in coronal section as this plane allows for the best direct comparison to the juvenile teeth in Fig. 4. In conjunction with its position relative to the teeth, it is tentatively identified here as a remodeling zone (*sensu Budney, 2004* Figs. 5B, 6A). In coronal section (Figs. 6A, 6C) the remodeling zone is clearly identifiable in cross-polarized light, and the distribution of dental tissues and bone is markedly different than that previously described in the juvenile specimen.

In the adult specimen, the anterior acrodont dentition was either completely worn away, or all that remained was a dentine fragment (Fig. 5). The most posterior dentition retained more comparatively more dentine and often with a lingual bias (Fig. 6B), but this was only discernable in thin section. The anterior acrodont teeth in the adult had no evidence of enamel, while the posterior most teeth were the only ones that maintained some enamel, although it is only present on the lingual side. Within the pulp cavity of the adult, there often remains evidence of vasculature (Fig. 6) although it is greatly reduced to the center of the tooth. The overall shape of the tooth is markedly different to that reported in the juvenile, and the external appearance is acrodont like that described in literature (e.g., *Budney, 2004*).

## DISCUSSION

Tooth replacement has been a topic of great interest in recent years in groups ranging from fish to tetrapods and across broad time scales. There have been several studies documenting the presence of replacement patterns in relation to dental development (e.g., *Westergaard & Ferguson, 1990*; *Richman & Handrigan, 2011*; *LeBlanc & Reisz, 2015*). However, there has been relatively little work on the anomalous squamates that completely cease tooth replacement, the acrodontians. Other than mammals, most fish (but see ratfish, e.g., *Huber, Dean & Summers, 2008*), amphibians, reptiles and non-mammalian synapsids have continuous tooth replacement through life, making acrodontian squamates an anomaly amongst toothed vertebrates. The lack of replacement comes with a set of challenges, two of which are addressed here. The first challenge is how does the jaw grow whilst having permanently ankylosed dentition and whilst maintaining occlusion with the maxillary dentition? The second challenge is how to maintain a single set of functional teeth through the lifetime of an animal, which essentially is a problem of combating or adapting to
wear. These two obstacles are also faced by mammals, which have one primary set of teeth throughout most of their life. In mammals, the issue of maintaining occlusion appears to be solved by maintaining a ligamentous tooth attachment, which allows for the teeth to remain mobile as the mandible grows and remodels (*Lumsden & Osborn, 1977*; *LeBlanc & Reisz, 2013*). The issue of wear is at least partially addressed by having much thicker prismatic enamel than that found in most reptiles (*Dauphin & Williams, 2008*; *Kieser et al., 2009*). However, the question remains how reptiles that permanently ankylose their dentition to the jaw and those with reptilian enamel adapt to growth and wear.

## Wear adaptations

This study found that the unworn teeth on the mandible of the juvenile agamid *P. vitticeps* have a uniform layer of enamel on both the lingual and labial sides, and the enamel is relatively thick in comparison to polyphyodont reptiles such as crocodilians. The thick enamel also seems to occur in two other acrodontians, *Uromastyx* enamel has been documented as 'thickened' (*Cooper & Poole, 2009*; *Throckmorton, 1979*), and chameleons' enamel appears thick in recent studies but is not explicitly commented on (*Buchtová et al., 2013*; *Dosedělová et al., 2016*). Furthermore, *Uromastyx*, a herbivorous acrodontian, has been reported to have thickened prismatic enamel (*Throckmorton, 1979*), similar to mammalian enamel. This shows that thickening of the enamel is a convergent adaption against wear in both squamates and in mammals.

Enamel thickening is not the only adaptation that acrodontians appear to have evolved in order to combat wear. Previous studies of chameleons and *Uromastyx* have shown that their pulp cavities were infilled with 'mineralized tissue' (*Dosedělová et al., 2016*) or 'bone' (*Throckmorton, 1979*). The likely purpose of this infilling is to prevent the pulp cavity from being exposed as the external surface of the tooth is worn away; both aforementioned studies also reported the disappearance or significant restriction of vasculature that initially supplied the dentition in early ontogeny. *Dosedělová et al. (2016)* also showed increased mineralization in the bone underlying the tooth-bone junction in chameleons, and *Throckmorton (1979)* found that in *Uromastyx*, the bone below the teeth had become more compact in appearance rather than cancellous. These findings are quite comparable with one another, even though *Uromastyx* and chameleons are on two disparate branches of Acrodonta (*Pyron, Burbrink & Wiens, 2013*). These results then beg the question: is this pattern more widespread across acrodontians?

This study found comparable results in *Pogona vitticeps*. The pulp cavity is also greatly diminished through ontogeny but is not completely 'obliterated,' as seen in *Uromastyx*. The infilling of the pulp cavity in *P. vitticeps* is done through a combination of dentine, and the bone. The progressive infilling of dentine is a normal development for teeth, as odontoblasts continue to sequentially deposit dentine post- eruption. What is abnormal is the bone growing into the pulp cavity and the resorption of the dentine root. In some of the sampled sections of *P. vitticeps*, the teeth fail to show the extent of bone infilling reported in *Uromastyx*, as the matrix that infills the pulp cavity does not seem to have any of the cellular spaces reported in *Uromastyx*. However, in other sections of the same individuals, the teeth experience a lot of remodeling and infilling, often with bone growing over the
dentition. This is identified as 'bone' rather than 'mineralized material' as mentioned by (*Dosedělová et al., 2016*) based on the amount of osteocyte lacunae present as well as the presence of osteons. Together the bone and dentine restrict the pulp cavity to such an extent that the tooth can then be worn away without the risk of exposing the vulnerable vasculature and nerves that are within the pulp cavity. Furthermore, the bone making up the dentary is not vascularized and sequentially also becomes more infilled with bone, making compact bone through ontogeny; this allows for the jaw to be worn in the form of distinct mandibular wear facets.

The amount of wear on the adult dentition of *Pogona vitticeps* changes the external morphology significantly. This is common in ungulate mammals (*Fortelius & Solounias, 2000*; *Kaiser et al., 2013*) and in herbivorous fossil reptiles (e.g., *Reisz, 2006*) but has not been reported in other squamates. The dentine and enamel on the labial side of the mandible are worn in the older specimens of *P. vitticeps*, which is compatible with the findings in *Uromastyx* (*Throckmorton, 1979*), where depending on the tooth position, the enamel and dentine were either completely worn away or were worn to such a degree that the lingual side of the tooth retained a much thicker layer of these two tissues. This pattern is likely caused by the combination of extensive wear from feeding and passive occlusion with the maxillary dentition, as well as the complete lack of tooth replacement. Interestingly, the vasculature that leads to the pulp cavities is still present in the adult specimens of *P. vitticeps* indicating that even the most worn teeth probably remained viable through the heavy wear process and likely closed the pulp cavity progressively until they were worn away. This phenomena has also been observed in hadrosaurs, which progressively infill their dentition with dentine as it nears the occlusal surface in their dental batteries and then wear away the entire tooth (*LeBlanc et al., 2016b*; *Bramble et al., 2017*).

## Maintaining occlusion

Continuous growth of the mandible throughout ontogeny is concurrent with the increased size of the skull. In most polyphyodont taxa, this is not problematic, as constant tooth replacement adjusts for increase in tooth size as well as possible migration (e.g., *Haridy, LeBlanc & Reisz, 2018*); even in mammals, after the aquasition of their permanent set of dentition, a maintained ligamentous attachment of teeth allows for migration and remodeling. However, for acrodont squamates that have ankylosed monophyodont dentition, this becomes problematic—the question becomes how occlusion is maintained as the jaw grows. Through examination of 37 specimens and histological sections, this study of *P. vitticeps* has shown that the growth of the jaw and the initiation of additional tooth development are decoupled processes. This is evident in the juvenile specimen in which the dentition is growing into the coronoid process (Figs. 3B, 3E, 3F), where the youngest un-erupted tooth is resorbing the ventral portion of the coronoid process in order to continue developing. This indicates that in the early stages of ontogeny, tooth development likely happens at a rate faster than dentary growth. This is reinforced by comparing the juvenile specimen (103 mm SV length), which has 16 tooth positions, with the adult, which is more than twice as long (222 mm SV length), only has 17 tooth positions. This supports the hypothesis that attaining the maximum number of teeth occurs

relatively early in ontogeny, well before attaining maximum adult size. Tooth development eventually slows and likely stops early in ontogeny, while the various jaw elements continue to grow. The mandible of *P. vitticeps* undergoes many changes through ontogeny, but there is little distortion of the tooth row during growth. This is likely achieved by appositional bone growth of the jaw, with deposition of parallel-fibered bone both internally and externally to achieve the increase in internal ossification, width and length that is seen in the adult and without remodeling or migration of the ankylosed dentition. Through the histological results of this study, which show little bone remodeling, and external observation of 37 specimens (see Table S1), it is hypothesized that occlusion is maintained through the allometric growth of the various portions of the jaw elements. The dentary and its corresponding dentition are the first to reach adult size, while the other jaw elements continue to grow through ontogeny, gradually increasing in robustness but not interfering with occlusion.

## Acrodont implantation as a result of ontogenetic remodeling

Acrodont implantation is defined ambiguously and inconsistently in literature. The mode of implantation has been defined as attachment to the 'edge' of a jaw (*Peyer, 1968*), as teeth ankylosed to the 'apex' of the jaw by cement (*Edmund, 1969*), or as the fusion of dentition to the 'margin' of the jaw (*Motani, 1997*). Eventually, the lack of replacement also became a character of acrodont tooth implantation (*Zaher & Rippel, 1999*), as the condition is present in acrodont squamates. Implantation categories are pervasive in literature and are often used as phylogenetic characters, but these definitions are problematic and tend to imply an evolutionary progression of tooth implantation from simple to more complex (see full implantation review in: *Budney, Caldwell & Albino, 2006*). In reality, the traditional categories of acrodonty, pleurodonty, and thecodonty are, at best, descriptive terms (*Estes & Charles, 1988*; Fig. 7A). *P. vitticeps* and likely other agamids change implantation categories through ontogeny, and therefore, they are a good representation as to why implantation categories should only be used descriptively. The teeth in the juvenile *P. vitticeps* are unequal in shape, with the lingual side being much longer than that of the labial side (Fig. 4), and attachment mostly occurring on the labial side, which are characteristics of pleurodont implantation. Traditionally in literature, the anatomy present in adult individuals of various taxa is described; this likely explains why acrodont squamates got their namesake since in their adult stage, the teeth do appear acrodont. This study reveals how implantation may appear to be acrodont in adulthood through a combination of factors (Fig. 7); (1) the gradual infilling of teeth with dentine that allows for progressive wear (2) resorption of the dentine by odontoclasts Figs. 4C and 4D, (3) growth on the dentary that infills part of the pulp cavity with woven bone and that reduces the trabecular structure of the dentary bone, (4) the continued remodeling of the dentary which removes traces of buried dentine and the previous pleurodont dentition. Tooth wear then further changes the crown morphology the maxillary dentition occludes with the mandibular dentition wearing away the labial side of the mandibular dentition and wearing away part of the dentary creating wear facets. This progression model shows how teeth that were originally pleurodont in implantation can appear acrodont and have an entirely different morphology due to a combination of dentine

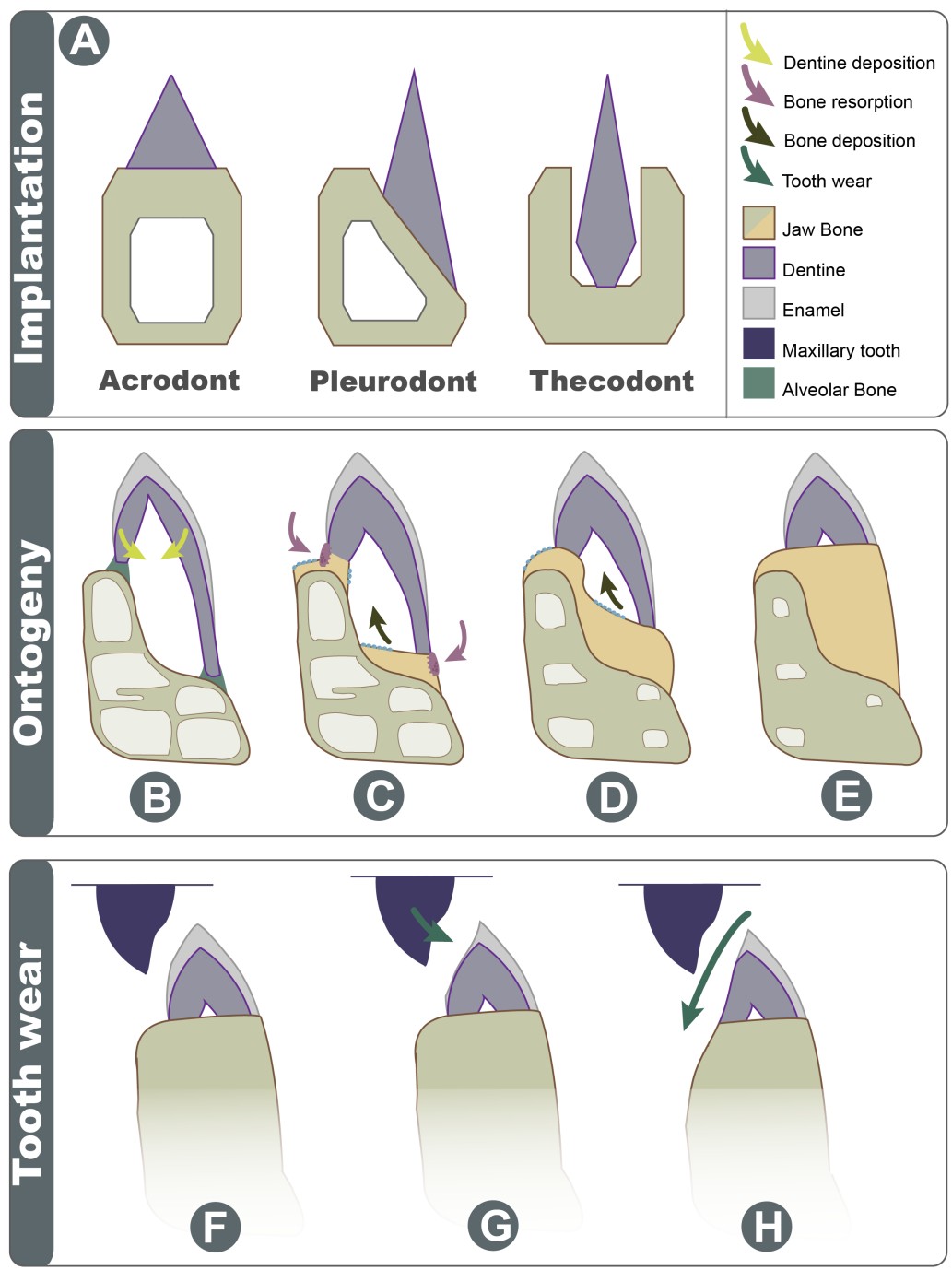

**Figure 7** Schematic explanation of implantation changes through ontogeny and tooth wear. (A) Three basic traditional implantation categories, without attachment tissue illustrated; (B) pleurodont impanated tooth is attached via alveolar bone, yellow arrows show direction of dentine deposition; (C) purple arrows show direction of dentine resorption by odontoclasts, blue arrow shows direction of bone deposition by odontoblasts; (D) blue arrow shows further bone deposition, trabecula in the dentary is reduced; (E) the tooth appears implanted at the apex of the dentary; (F) Shows the position of maxillary tooth; (G) the enamel is worn on the labial side; (H) with advanced wear, the enamel, dentine, and dentary bone are worn on the labial side.

infilling, bone remodeling, and tooth wear. It is difficult to ascertain without experimental testing if these secondary remodeling steps would occur in any pleurodont squamate taxa but are suppressed by tooth replacement, or if these are secondarily acquired adaptations in monophyodont taxa to combat wear. Therefore, at least in the agamid *P. vitticeps*, the dentition changes from pleurodont to acrodont in appearance through ontogeny. This indicates that we should not use implantation categories as phylogenetic characters if the ontogenetic stage of the specimen is unknown, as is the case in many fossil taxa.

## CONCLUSION

The dentition of acrodontians has recently been an area of interest in the context of dental development and implantation, as this group is known for its mode of implantation and monophyodonty. The latter is an anomalous occurrence among tetrapods, as intermittent to continuous replacement of teeth is the primitive condition for toothed vertebrates, including most extant fish, amphibians, and amniotes. This lack of replacement has become associated with acrodonty, resulting in monophyodonty becoming a formal characteristic of acrodont implantation in squamates, and previous authors have implied that acrodont implantation is the cause of monophyodonty. However, this study shows that at least some modern acrodont reptiles do not initially have acrodont implantation early in ontogeny, and there is a distinct ontogenetic change in the morphology and implantation of dentition of the agamid *P. vitticeps.* The youngest teeth in juvenile specimens are pleurodont in implantation, with a greater lingual contribution of tooth tissues and an attachment biased towards the labial side. These tissues are secondarily remodeled through a step-wise process of: (1) resorption of dentine, (2) deposition of bone and dentine, (3) wear of the tooth surface, and (4) wear of the jaw bone proper, effectively changing the morphology. These processes give *P. vitticeps* the appearance of acrodont dentition in adulthood. This is an important distinction to make as it signifies that this acrodont squamate, and likely other acrodont reptiles, do not develop acrodont teeth, but rather develop pleurodont teeth like the vast majority of squamates, making their teeth inherently pleurodont, and secondarily acrodont. Therefore, the wear adaptations, remodeling, and dental wear come together to give the appearance of an acrodont mode of implantation. The wear adaptations documented in this study likely evolved due to the lack of replacement. Importantly the monophyodont condition causes the acrodont condition, rather than acrodonty causing monophyodonty, as has been implied in literature.

This raises the question as to how we should code acrodont dentition in squamate phylogenies; is it true acrodonty if this implantation mode is only achieved through secondary remodeling of the teeth and dentary? Finally, another important consideration, and a direction of future studies, is that if all acrodontians share these ontogenetic changes and wear adaptations, then what is the significance of the convergence seen between acrodontian squamates and the rhynchocephalian *Sphenodon*, which is also reported to have acrodont dentition?

## ACKNOWLEDGEMENTS

Incredible thanks go to Robert R. Reisz and Aaron H. Leblanc who brought this issue to light and encouraged the development of this study. I would like to thank Diane Scott for photographs and SEM images and bearded dragon husbandry advice. Thanks to Bryan Gee for his everlasting patience and editing of this manuscript into presentable form. I would like to thank Kevin Seymour of the Royal Ontario museum (ROM) for assistance in locating appropriate specimens for this study and to thank David Evans for allowing the use of the ROM histology laboratory and reading early versions of this manuscript. Thanks goes to Megan Whitney and Kelsey Jenkins for their helpful reviews which improved this manuscript. Lastly, I would like to acknowledge Osiris the bearded dragon whose presence in the lab was essential to the direction of this study.

### Funding

The authors received no funding for this work.

### Competing Interests

The authors declare there are no competing interests.

### Author Contributions

- Yara Haridy conceived and designed the experiments, performed the experiments, analyzed the data, contributed reagents/materials/analysis tools, prepared figures and/or tables, authored or reviewed drafts of the paper, approved the final draft, composing figures, obtaining specimens.

### Data Availability

Raw data is available in the Supplemental Materials.

### Supplemental Information

Supplemental information for this article can be found online at http://dx.doi.org/10.7717/peerj.5923#supplemental-information.

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
