# Peer review of "Histological analysis of post-eruption tooth wear adaptations, and ontogenetic changes in tooth implantation in the acrodontan squamate Pogona vitticeps"

_PeerJ, doi:10.7717/peerj.5923_

## Round 0.1 · original submission · Minor Revisions

Dear Yara,

Your manuscript needs minor revisions before it can be resubmitted. Please consider the comments of the reviewers in particular in respect to redundancies. And for readers not so knowleabel about teeth I would like to have a better differentiation and definition of "tooth attachment" and "tooth implantation" as you note that the difference is importance; but I think it is not clear enough in the manuscript.

Both reviewers have provided annotated manuscripts

·

Basic reporting

I think this is a great paper; interesting findings. I would be glad to see in published in the near future.

I have a few areas to suggest improvement that I've noted throughout the document. They cover two areas:
1. Citations - This is mostly in the introduction, but a few other areas as well. When you say something is "common," "pervasive," or "seen frequently," that implies there is probably a large body of publications supporting that idea, so there should be multiple (and possibly older than 1990s usually) works cited for that idea.
2. Grammar - I tend to be a little overenthusiastic about grammar, so feel free to ignore my comments on grammar if you feel strongly about it. However, I do believe there are a few areas where you use therefor instead of therefore. Also, I strongly suggest not to use "as" or "since" when you really mean "because." I'm sure the reader knows what you mean, but it's a little pet peeve of mine because those two words are not direct synonyms of because. I've also pointed out a few run-on sentences.

You'll also see this in my comments, but trogonophonid amphisbaenians are also acrodont and are squamates, but they do not belong to Acrodonta. That should be noted in your intro (and I think I may have pointed it out again in your discussion?). There's a few publications on this, but there are also a few CT scans available to view on digimorph.org.

Experimental design

no comment

Validity of the findings

no comment

Additional comments

Good findings! I thought it was not only interesting, but considering that the acrodont teeth of agamids have long been thought to be life-long acrodonts, these are also novel findings. I think this work with encourage future research on the topic; it would be interesting to see if a similar pattern is seen in other acrodontians, trogonophonids, and rhynchocephalians.

·

Basic reporting

• It is important that the author sets up that the definition of acrodonty is ambiguous at best, however, this reviewer believes it is important to establish what definition of acrodonty the author is ascribing to for the remainder of the manuscript. That is never established in the introduction (lines 86-99).

• Better, close-up images of the wear facets on the jaws and teeth would be helpful.

• When using “wear facets” it would be clarifying to specifically refer to either the tooth or jaw wear facets so the read knows which is being discussed.

• Line 251: a reversal line is mentioned and it would be helpful to the reader if that was labeled in a figure.

• Line 263: Also important to note that the location (i.e. deep in the jaw) also indicates that it is a developing tooth.

• Line 281: The wear facets are not really well figured as mentioned in this sentence.

• Line 285: Label pseudo-pedicels in figures. Additionally, a comparison between the ‘bony pedicels’ of the chameleon and the pseudo-pedicels of the Pogona vitticeps.

• A more detailed description of the woven bone would be beneficial. Vascularity? Sharpey’s fibers? Osteocytes? A reorganization of this paragraph would clarify important points. Consider first just describing this tissue, then how the location and tissue-type suggests remodeling (also, please provide a citation for why this tissue suggests remodeling), and then discuss how remodeling is further evidenced by the geometry of implantation.

• There are good descriptions of the enamel on the lingual and labial sides of the teeth but not the mesial or distal sides. A description would be informative and if not available, indication as such would be helpful.

• It looks like all if not most of the figures in the manuscript are mislabeled. A review by the author of all of these numbers is necessary.


• Figure labeling is not consistent in a few ways. First, double check abbreviations described in figure legend and those used in images. Second, in this histology images, it is inconsistent whether or not the external view is labeled as A or left blank. It seems most helpful to label as A (which is done in the figure legends, but not on figures themselves) and then shift the following histological images to B, C, D, etc.

Experimental design

• If it is known, it would be interesting to know if these museum specimens are zoo, captive, or wild specimens.

• Citation and justification for recognizing the differences in the jaws as ontogenetic rather than different species would add robustness to this analysis.

Validity of the findings

• The longitudinal sections do not apparently reveal any enamel. Does that mean that there is no enamel on the mesial and distal surfaces? If so that is really interesting and worth mentioning. If it is simply due to sectioning bias that is also worth mentioning.

• Discussion around the use of acrodonty as a phylogenetic character is well-written and very extremely interesting and important.

Additional comments

Overall I believe this to be a significant and extremely important piece of research that absolutely merits publication. This paper is most significant in its identifying the more nuanced the tooth implantation type of Pogona vitticeps and importantly provides evidence for and calls for a paradigm shift in the use of a descriptive feature like tooth implantation as a synapomorphy for clade definitions. Some minor additions to the photos and descriptions are suggested as well as, some organizational and persistent labeling issues that should be addressed. This reviewer did feel that there were several times where the research intent questions and general introductory material were repeated so please take care to make sure to cut back on redundancy.

---

## Round 0.2 · accepted · Accept

The comments of the reviewers were all taken into account and the manuscript was revised accordingly and therefore can be published.

There are a couple of minor edits to address in production:

Zahn and Reippel insteaed of Rieppel (L 51) and a missing space in l 52

#